# Exploring factors of e-waste recycling intention: The case of generation Y

**Muhammad Yaseen Bhutto** [ID]¹, **Aušra Rūtelionė**¹*, **Beata Šeinauskienė**¹, **Myriam Ertz**²

**1** School of Economics and Business, Kaunas University of Technology, Kaunas, Lithuania, **2** Labo NFC, Department of Economics and Administrative Sciences, University of Quebec in Chicoutimi, Saguenay, Canada

* ausra.rutelione@ktu.lt

**Data Availability Statement:** The datasets used during the current study cannot be shared due to respondents' privacy concerns. As such, participants were assured that data will only be used for research purpose and will not be shared

## Abstract

The seriousness of the e-waste crisis stems from the fact that consumers do not participate much in ensuring the proper disposal of electronic materials. In this context, millennials are the largest segment of consumers of electronic products who are not yet motivated to get sustainably rid of them. However, to inspire consumers to recycle e-waste, it is necessary to investigate consumers' behavioral intentions towards e-waste thoroughly. This study integrates the theory of planned behavior, social influence theory, and personality traits to examine how consumers gauge their choice to recycle e-waste. Data were collected from randomly surveying 300 Lithuanians through a structured questionnaire. Using the PLS-SEM approach, results show that attitude, subjective norms, and perceived behavioral control significantly influence consumers' e-waste recycling intention. Regarding personality traits, only openness to experience significantly affects consumers' e-waste recycling intention. In contrast, other traits such as agreeableness, conscientiousness, extraversion, and neuroticism have a non-significant influence on consumers' e-waste recycling intention. In addition, normative and informational social influence affects consumers' e-waste recycling intention. The current study advances our understanding of e-waste recycling behavior by examining how TPB, personality factors, and social influence theory influence intentions. It provides valuable insights for policymakers and marketers on understanding and encouraging the e-waste behavior of Lithuanian Y-generation consumers.

## 1. Introduction

Natural resources have been depleted and over-consumed due to fast economic expansion and industrial advancement, particularly over the previous two decades. As a result, significant environmental issues today exist, including water and air pollution, land degradation, forest destruction, and climate change [1]. One of the factors contributing to this dilemma is the (over)consumption of electronic gadgets as more and more individuals join the global information community and the digital market [2, 3]. Furthermore, more individuals use several electronic devices and products like computers, smartphones, and other electronic equipment that have shorter lifespans [4, 5]. As a result of this phenomenon, WEEE, also known as e-waste, or waste of electrical and electronic equipment, is sharply increasing [6]. Waste

to third party for any reason. Therefore, data cannot be publicly shared. However, those who are interested in data sets can request specially to Dr. Aušra Rūtelionė or Ethics Committee of Kaunas University of Technology on tyrimu.etika@ktu.lt.

**Funding:** This research is part of the project that has received funding from European Regional Development Fund (project No. 01.2.2-LMT-K-718-03-0104 under grant agreement with the Research Council of Lithuania (LMTLT).

**Competing interests:** The authors have declared that no competing interests exist.

management is an important issue that negatively affects sustainable health care and reduces environmental quality [7]. Worldwide, e-waste generation grew three times faster than global population growth [6]. By 2030, it is anticipated that the quantity of e-waste generated worldwide will exceed 74 Mt. A number of electronic items, including smartphones, contribute to the majority of electronic waste. In addition, due to the coronavirus pandemic, the number of smartphone users worldwide has sharply increased. During the pandemic, people worked and studied online, which led to a more intense usage of electronic communication devices, particularly smartphones. The usage rates are 70% for smartphones, 32% for personal computers, and 40% for laptops. Due to the chemicals produced during ignition, the disposal of e-waste also contributes to climate change. Metals like copper (Cu), aluminum (Al), and iron (Fe), which are found in electronics and become airborne when burned [8].

Europe's well-developed e-waste infrastructure allows private firms to gather e-waste from stores and communities, recover recyclable portions from the e-waste collected, and dispose of leftovers legally and environmentally responsibly [9]. As a result, the e-waste recycling rates are highest in Europe compared to Asia and South America [10]. Considering European indicators, according to Eurostat [11], e-waste recycling rates were below 50% in most European countries, except Estonia, Iceland, Hungary, and Austria, which exceeded the 50% recycling threshold for e-waste; Liechtenstein recycled more than 80%. In contrast, the lowest e-waste recycling rates were recorded in Lithuania, Iceland, Poland and Greece.

The extensive research on e-waste recycling adoption behavior has been documented in several articles in different countries, such as China [12–14], the USA [15, 16], Norway [17, 18], Greece [19], Italy [20], New Zealand [21], South Korea [22], France [23], Finland [24], Germany [25], Australia [26], Romania [27], Japan [28]. Although the Baltic States have a low recycling rate, there is less evidence on the determinants of e-waste in those countries. Besides one of them, Lithuania is facing severe problems regarding e-waste management [29]. Lithuania [30] has a well-developed e-waste collection network; State institutions have been training the population for many years and explaining how important it is to separate electronic waste and dispose of it responsibly; for more than a decade, collecting bulky electronic waste from households free of charge. However, it turns out that a significant part of the population still does not practically accept e-waste. Recently, Lithuania's Ministry of Environment of the Republic [31] made a case about general waste management; the long-term objectives lay the groundwork for waste management capacity planning, where at least 65% of waste should be recycled or reused. Similarly, scholars pointed out research efforts required to examine the progress of e-waste adoption behavior in Lithuania [29, 32]. For this reason and access to data, this country is taken as the main case study in this paper. To ensure recycling e-waste can benefit the environment, it is necessary to understand consumer behavior, particularly regarding e-waste. So far, there is a lack of research to identify consumers' e-waste recycling intention in a Baltic country such as Lithuania.

Most e-waste behavior research has been conducted on how attitudes and beliefs affect people's e-waste behavior to understand better adopting e-waste recycling [27, 32–34]. Similarly, grounded in the theory of planned behavior (TPB), the literature mostly tackles consumers' intentions toward e-waste; researchers have incorporated several variables in the TPB, including individual responsibility and awareness of consequences [33], habits [34], past recycling experiences [35], demographics [36], government initiative and consumer knowledge [37], or consumer values such as altruistic, biospheric, and egoistic values [38]. Personality traits refer to inherited distinctive patterns of thoughts, feelings, and behaviors [39]. Personality traits reflect how a person makes decisions [40] and may lead to an inherited basis [41], which may explain individual differences in e-waste recycling intentions and behaviors. Duong [42] confirmed that the Big Five's personality traits have recently

been used in literature as precursors to ecological behavior. Few empirical evidence linking personality traits to pro-environmental behaviors such as green investment [43], green purchase [42], sustainable transport [44], household energy conservation [45], energy-saving behavior [46], and green information technologies [47]. However, the mechanism of how personality traits influence e-waste recycling behavior is still underexplored and remains a significant research gap.

Some researchers have recently examined environmental consumption through a social influence theory perspective [48–50]. Social influence provides information and motivation to individuals to develop and accept new behaviors [51]. [52] explained the two major factors of social influence theory. The first is known as informational social influence (ISI), which is about receiving information from others as proof of the reality of something. On the other hand, normative social influence (NSI) is about enforcing the expectations of others in the group, sustaining harmony, and valuing the positive evaluations of others [53]. Empirical data supports the claim that social influence can be a potent force in encouraging sustainable behavior, such as the intention to recycle plastic, sustainable households, towel reuse, and energy conservation [48, 51, 54, 55]. Nevertheless, the NSI and ISI were infrequently used as predictors of sustainable consumption behavior in the literature, according to Hameed et al. [48]. However, no work has been done to investigate the impact of NSI and ISI on consumers' e-waste recycling intention, although extensive research suggests that NSI and ISI are important factors in encouraging consumers to engage in sustainable consumption [48, 51, 54, 55].

Researchers have cited generational effects and dissimilarities in sustainable consumption [56–59]. Generation Y (Gen Y), also known as "Millennials," is a generational group born between 1981 and 1999. It has become a promising consumer segment with significant purchasing power and garnered much interest from researchers [60]. This consumer group differs from prior generations, which has generated much research interest [61]. In addition, marketers predict that millennials have a $200 billion purchasing power [62]. Although interested in acting responsibly, Generation Y individuals are more financially constrained than prior generations [63]. On the other hand, members of this generation are well-educated and better understand sustainable development [64]. Due to their extensive exposure to technology, millennials are long-term electronic users and have tremendous consumption potential [65]. However, previous research has shown that research on millennials' propensity to recycle e-waste has not been widely understood [66, 67].

To address this issue, the study proposes a conceptual framework based on the two main factors of TPB, personality traits, and social influence theory (i.e., normative social influence and informational social influence) to investigate the e-waste intentions of Lithuanian millennials. Millennials stand out because they are the most influential consumer class in the global market. This class shows great concern for the environment [64] but is often neglected in environmental studies [68, 69]. To promote socially responsible e-waste management and guide future behavior, it is important to understand what influences millennials' e-waste recycling intentions. A better understanding of such intentions will help guide millennials from informal disposal to sustainable recycling methods. This facilitates the preservation of valuable resources of e-waste and reduces its harmful environmental effects in the future by addressing the following questions in the study;

1. What drives Millennials' intentions toward e-waste recycling?

2. What is the impact of integrating TPB theory, personality traits, and factors like normative social influence (NSI) and informational social influence (ISI) to predict Millennials' intention toward e-waste?

The novelty of this study is that it is the first attempt of its kind to predict Lithuanian millennial' behavioral intentions to adopt e-waste recycling and integrate TPB, personality traits, and the two main factors of social influence theory (i.e., normative social influence and information society influence), to identify the factors that influence their intentions. This study recommends managers, regulators, and operators promote e-waste recovery and sustainable e-waste management practices.

## 2. Literature review

### Theoretical framework

E-waste recycling is a typical eco-friendly behavior, and it is necessary to promote it as it is the cause of many environmental problems [70]. Particularly today, e-waste pollution and resource scarcity are serious problems restraining sustainable development [71]. To stimulate individual recycling behavior for e-waste, one needs to know the general regulation of this behavior. Therefore, it is essential to explain the factors that drive people to recycle e-waste [72]. To stimulate individual recycling behavior for e-waste, one must know the general regulation of this behavior. Unfortunately, it is difficult to explain why people engage in e-waste recycling [72]. To understand e-waste recycling behavior, the Theory of Planned Behavior (TPB) has been widely used as an excellent model to explain recycling behavior [34] and is considered the most reliable and authentic theory to investigate sustainable behavior [73, 74]. Recently, TPB has been used to explore the e-waste recycling behavior of household residents, youth, students, teachers, and other groups, including general household waste, electronic waste (e-waste), and construction waste [26, 27, 33, 36, 37, 75]. The theory enables researchers to expand the theory and better comprehend human behavior in a given context by incorporating additional factors [76]. Therefore, this study combines the theory of planned behavior (TBP) [76] as well as normative and informational factors of social influence theory (SIT) [52] and Big Five personality traits, recognized as conscientiousness, neuroticism, agreeableness, extraversion, and openness to experience [77]. This theoretical assemblage contributes to a better understanding of the factors driving consumer behavior toward adopting e-waste recycling.

### Attitude, subjective norms, perceived behavioral control and consumers' e-waste recycling intention

Attitude expresses an evaluative response to a particular case, whether favorable or adverse. It is generally a predetermined responsive state associated with a particular object, subject, or entity [34]. Ajzen [76] defined attitude as a person's opinion on engaging in a specific behavior, whether they think it's favorable or unfavorable. Attitude has also been recognized as strong interpreters of pro-environmental behavior because of their capability to tolerate doubts and risks arising from adopting a decision [78]. In the pro-environmental literature, Soomro et al. [79] documented that attitude denotes the positive or negative assessment of behavior toward recycling intention. Sabbir et al. [37] discovered that a positive attitude toward e-waste recycling influences e-waste recycling behavior. Numerous pieces of research have revealed a good relationship between attitude and intention to recycle e-waste [34, 35, 65, 66]. As a result, the subsequent hypothesis can be assumed:

*H1*: *Attitude positively affects consumers' e-waste recycling intention.*

The subjective norm idea represents an individual's impression of social pressure to perform or refrain from performing a specific conduct [61]. It is a process wherein the beliefs of a reference group or, more specifically, significant persons, such as family, friends, and the

community, affect an individual's observations, opinions, and feelings [34]. The strong influence of subjective norms on pro-environmental intention has been shown in preceding studies [73, 80, 81]. Soomro et al. [64] studied psychological factors influencing solid waste recycling intentions. They contended that individual social norms could significantly increase the intention to recycle solid waste. Nguyen et al. [35] investigated behavioral intentions to recycle e-waste. According to the findings, subjective norms are the most important forecaster of consumer intentions to recycle e-waste. Similarly, Aboelmaged [34] used the TPB framework to investigate the drivers of behavioral intentions to recycle e-waste and discovered that subjective norms are the most important predictors of consumers' intention to recycle e-waste. The following hypothesis can be presented based on the preceding discussion:

*H2*: *Subjective norms positively affect consumers' e-waste recycling intention.*

Perceived behavioral control is "people's perceptions of their ability to perform a particular behavior" [76]. It relates to a person's sense of ease or difficulty in a specific task [82]. Kianpour et al. [83] revealed that behavioral control strongly predicts household users' intention to recycle or reprocess outdated household or electronic devices every week. Besides, past studies showed that perceived behavioral control positively influences recycling intention and behavior [79, 84]. Similarly, Aboelmaged [34] mentioned PBC is an essential determinant of e-waste recycling intentions. In conclusion, based on our review of the literature, we propose the following:

*H3*: *Perceived behavioral control positively affects consumers' e-waste recycling intention.*

## Personality traits and consumers' e-waste recycling intention

Scholars characterize the Big Five personality qualities as agreeableness, extraversion, conscientiousness, neuroticism, and openness to experience [85]. However, only limited studies have inspected the links between personality traits and environmental consumption [86]. Interestingly, several researchers have suggested Big Five personality traits are associated with ecological consumption [42, 87, 88]. Consequently, they might also be related to pro-environmental behavior, such as recycling e-waste. Agreeableness is the desire for generosity, compassion, social harmony, and the motivation to interact with others [47]. Agreeable people are habitually comfortable, pleasant, helpful, cooperative, and enjoy assisting others. They can also pay closer attention to others' needs and the natural environment [89, 90]. Furthermore, agreeable people tend to show more outstanding environmental friendliness because it is publically accepted as virtuous and concurs to shape someone's "good citizen" image [86]. Previous studies have revealed that agreeableness correlates with pro-environmental behavior, but the results have been contradictory. Although some studies have reported that agreeableness negatively influences pro-environmental behavior [87, 90], others found a positive relation between agreeableness and pro-environmental consumption [42, 89, 91]. Theoretical reasoning is nonetheless conducive to considering agreeableness as being positively related to e-waste recycling intention. Since such behavior is socially promoted, encouraged, and appreciated, it might be conducive to pleasing others. Yet, being pleasurable to others is an inclination that is deeply embedded in an agreeable personality. Therefore, based on this literature, we suggest that agreeableness enhances e-waste intention among millennials.

*H4*: *Agreeableness positively affects consumers' e-waste recycling intention.*

Conscientiousness is a quality that reflects a tendency toward self-discipline, a sense of duty, commitment, and devotion to rules and customs [92]. It is also associated with being more

attentive to the future by thinking about it more often [87]. Furthermore, conscientious people are likely to diagnose severe environmental problems more quickly because they have greater environmental interests [86] and are inclined to take applicable measures to guard the environment [93]. However, the results are mixed as well in the literature. Several studies have shown that conscientiousness is related to higher levels of pro-environmental engagement [87, 89, 94], whereas others have claimed that conscientiousness is not so closely associated with pro-environmental engagement [42, 95]. A recent study found conscientiousness positively influences smartphone recycling intention [96]. Hence, the subsequent hypothesis can be suggested:

*H5*: *Conscientiousness positively affects consumers' e-waste recycling intention.*

Extraversion is described as self-confidence, high talkativeness, pleasantness, and active participation in the community and society [45]. Extraverts are willing to support others and positively influence environmental behavior [94]. In addition, they are active, friendly, and relaxed when surrounded by large groups. Due to their sociable personality, they prefer large shared support networks and find opportunities for communication [97, 98]. According to Milfont and Sibley [89], extraversion is strongly related to environmental commitment. Post-materialistic values like subjective well-being and self-expression, in particular, are positively associated with extraversion [39], higher environmental considerations, and pro-environmental views. Duong [42], on the other hand, recently discovered no significant association between extraversion and intention to involve in ecologically friendly consumption. Existing research yields inconclusive results, implying that the association between extraversion and ecologically friendly behavior requires further examination. The following hypothesis is proposed in this study:

*H6*: *Extraversion positively affects consumers' e-waste recycling intention.*

Openness to experience does not just relate to an individual's need for information, creative abilities, and preference for novelty [45]. It is also associated with rich opinions and appreciation for different and uncommon experiences [87]. Previous research indicated that openness to experience is positively linked with pro-environmental behavior [73, 75, 86]. People with high openness are expected to engage in ecologically-friendly consumer behavior [42]. As a result, the following hypothesis is proposed in this study:

*H7*: *Openness to experience positively affects consumers' e-waste recycling intention.*

Neuroticism reflects the tendency to experience negative emotions, containing fear, anger, unease, and psychological distress [45]. According to Hirsh et al. [99], people with high levels of neuroticism are more concerned about negative consequences, and their environmental distress makes them afraid of the ecological damage that waste can cause. Numerous recent studies have found that neuroticism has a role in pro-environmental behavior [39, 73, 80]. Neuroticism influences pro-environmental behavior in a positive and significant way [88]. More research has indicated that neuroticism negative influence on pro-environmental behavior [40, 75, 83]. Therefore, the subsequent hypothesis can be suggested:

*H8*: *Neuroticism positively affects consumers' e-waste recycling intention.*

## Normative social influence, informational social influence, and consumers' e-waste recycling intention

According to social influence theory [89], it refers to the thoughts, attitudes, beliefs, and behaviors of persons who are affected by others. Social influence theory is appropriate for this study

since it has been extensively researched in several research fields, including social commerce, addictive behavior, and pro-environmental conduct [100]. Social influence is the intention to decide based on social pressure [101]. People inclined toward NSI conform with others [102]. They do this to escape punishment, earn rewards, or build close relationships with other group members [93]. NSI has previously been utilized in investigations of new information system acceptability [94, 95]. In addition, the concept has been used in the context of sustainable consumption [73, 103], including organic food [103], towel reuse [104], plastic waste recycling [48], and energy conservation [55], has provided positive results. People with higher levels of NSI also feel stressed about performing a behavior others want [105, 106]. Thus, it is hypothesized that:

H9: *Normative social influence (NSI) positively affects consumers' e-waste recycling intention.*

Informational social influence is a process by which people identify the successful experiences of their social group before deciding to adopt an innovation [107]. Informational social influence helps to exchange information and strengthen relationships between individuals and peers [100]. Several studies have sought to comprehend the impact of information social influence on consumption [100, 108, 109], including sustainable consumption [100, 110, 111]. Recently, Hameed et al. [48] found that social influence positively influences consumers' plastic waste recycling intentions. Individuals who get more information about e-waste recycling encourage to do the same. They feel like they are doing something good by recycling waste because others also believe in recycling [51]. Thus, it is hypothesized that:

H10: *Informational social influence (NSI) positively affects consumers' e-waste recycling intention.*

Based on the hypothesis, we intend a conceptual model presented in Fig 1, which considers the influence of TPB, Social influence theory, and Personality traits' effects on e-waste recycling intention.

## 3. Methodology

### Data collection and survey instrument

In this study, we developed a pre-tested questionnaire and digitally distributed it to a Lithuanian population aged 25–41. Data were collected in April 2022. The study relied on an online professional research group of approximately 20,000 active panelists willing to participate in the survey. The organization providing the data uses computer-assisted interviewing (CAWI) to collect data and uses a pre-assessment method to capture respondents' attention. We calculated a two-tailed test to test the difference between the two proportions (initial and effective samples). Based on z-scores and considering all demographic information, the results showed that the two proportions are not significantly different. A survey was issued to 858 respondents to collect a representative sample using the random sampling technique, and 300 valid responses were received. Because it offers comparable data, this sampling method is chosen and differs significantly from traditional random sampling approaches.

Three hundred respondents made up the data set; 145 (48%) of them were female, 153 (52%) were male, and 2 (1%) were other. Age of the respondents ranged from 25 to 41. The respondents' ages ranged from 25 to 30 (N = 90), 31 to 35 (N = 106), and 36 to 41 (N = 104). The majority of respondents (N = 203) had a higher education (university) degree, followed by respondents with a college degree (N = 6), respondents with higher secondary and middle school degrees (N = 45), and respondents with elementary school degrees (N = 4). (See Table 1).

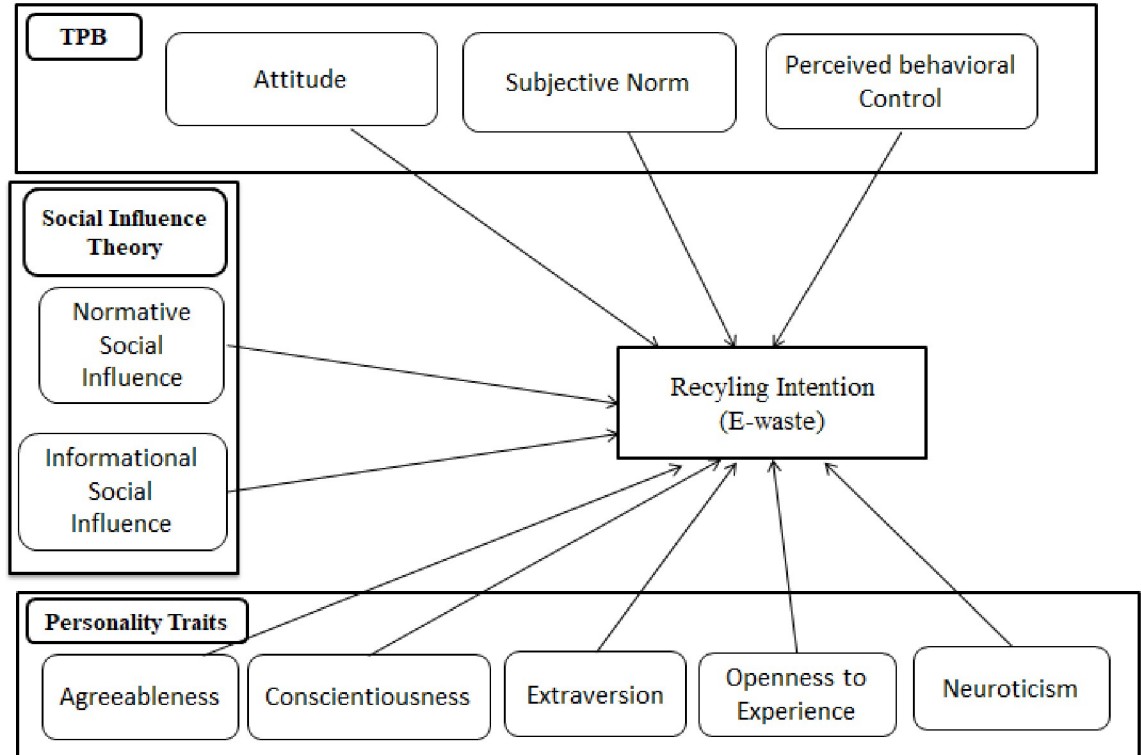

**Fig 1. Conceptual model.**

## Measurements

Items for measuring attitude, subjective norm, and perceived behavioral control are adapted from Kumar [33]. In order to measure e-waste recycling intention, four items were adapted from Nguyen et al. [35]. To measure the Big Five personality traits, Agreeableness,

**Table 1. Respondent.**

| Category | Frequency | Percentage |
|---|---|---|
| Gender | | |
| **Male** | 153 | 51% |
| **Female** | 145 | 48% |
| **Other** | 2 | 1% |
| **Don't want to answer** | 0 | 0% |
| Age | | |
| **25–30** | 90 | 30% |
| **31–35** | 106 | 35% |
| **36–41** | 104 | 35% |
| Education | | |
| **Primary** | 4 | 1% |
| **Middle** | 42 | 14% |
| **Higher Secondary** | 45 | 15% |
| **College** | 6 | 2% |
| **University** | 203 | 68% |

Conscientiousness, Extraversion, and Openness to Experience were adapted from Sun et al. [86] and took the neuroticism items from Duong [42]. Items for normative social influence were adapted from Taylor and Todd [112] as well as Hameed et al. [48], while information social influence items were adapted from Henningsen et al. [113]. All items were evaluated on a 5-point Likert scale, from strongly disagree (1) to strongly agree (5). S1 Appendix displays all the items used in the questionnaire.

## 4. Empirical results

This section discusses the PLS-SEM done with SmartPLS 3.0 to examine the conceptual model. The analysis was conducted in two stages. First, the measurement model was validated using SmartPLS to confirm its validity and reliability. Second, the structural model employs the two-step SmartPLS analysis technique to test the hypothesized linkages.

### Measurement model results

This study used partial least squares structural equation modeling (PLS-SEM) to estimate the measurement and structural models. PLS-SEM is a reliable method that works well for developing theories and doesn't need data standardization [105]. We used two ways. We began by evaluating the model's dependability and validity. The model was then analyzed in order to test hypotheses.

The composite reliability (CR) values for all constructs are greater than 0.70, indicating that the data were trustworthy and logically consistent [106]. CR values should be larger than 0. 70, and AVE values should be greater than 0. 50, to establish convergent validity [107]. Convergent validity determines that all constructs' cr and ave values were above the recommended cut-off levels, as shown in Table 2 and Fig 2. The only exception was extraversion with a CR valued below 0. 70, but since the ave was above. 50, internal consistency was ensured for the focal construct. A smaller subset of 32 items was retained in the entire model because their loadings were greater than 0. 55, the recommended standard suggested by Hair et al. [114] (Table 2) indicates that the measurement model was reliable and meaningful. In addition, according to the Hu and Bentler [115] standard, a model is considered well-fitted if its square root mean residual (SRMR) score is less than 0.09. This study saturated and estimated models have SRMR values of 0.069 and 0.071, respectively, indicating that they fit well.

The degree to which a construct is dissimilar to other constructs is called its discriminant validity [106]. According to Fornell and Larcker [116], discriminant validity is assured when the value of the square root of a single factor's AVE is greater than the sum of that factor's correlations with all other factors. The overall value of the square root of the AVEs is larger than the comparable value, demonstrating discriminant validity, as shown in Table 3.

The dimensions of the beta coefficients, the associated t-statistics, and the measure of $R^2$ for endogenous constructs are considered for structural model evaluation [109]. The bootstrapping method was used to determine the significance of the path coefficients. We also calculated the size effect ($f^2$) for each structural path with these parameters, as Hair et al. [114] suggested. $R^2$ used to evaluate the criterion for inner model fit [117]. $R^2$ value is the variation in endogenous constructs that is explained by exogenous constructs. The value of $R^2$ for the endogenous constructs e-waste recycling intention was 0.637 (Fig 2). More formally, the model accounts for more than half of the variance in the dependent variable's intent, which is relatively high, particularly in a social science context.

The results are presented in Table 4; according to the SEM findings, attitude (β = 0.150; t-value = 2.914; p-value 0.004), subjective norms (β = 0.147; t-value = 2.725; p-value 0.006), and perceived behavioral control (β = 0.141; t-value = 5.203; p-value 0.01) have a significant

**Table 2. Factors loading, CR, and AVE.**

| Constructs | Items | Loadings | CR | AVE |
|---|---|---|---|---|
| Attitude | | | 0.798 | 0.576 |
| | AT1 | 0.888 | | |
| | AT2 | 0.771 | | |
| | AT3 | 0.587 | | |
| Subjective Norms | | | 0.933 | 0.823 |
| | SN1 | 0.918 | | |
| | SN2 | 0.915 | | |
| | SN3 | 0.889 | | |
| Perceived Behavioral Control | | | 0.931 | 0.818 |
| | PBC1 | 0.882 | | |
| | PBC2 | 0.923 | | |
| | PBC3 | 0.908 | | |
| Normative Social Influence | | | 0.769 | 0.526 |
| | NSI1 | 0.690 | | |
| | NSI2 | 0.729 | | |
| | NSI3 | 0.755 | | |
| Informational Social Influence | | | 0.874 | 0.699 |
| | ISI1 | 0.869 | | |
| | ISI2 | 0.872 | | |
| Agreeableness | | | 0.863 | 0.679 |
| | AG1 | 0.750 | | |
| | AG2 | 0.854 | | |
| | AG3 | 0.863 | | |
| Conscientiousness | | | 0.812 | 0.598 |
| | C1 | 0.813 | | |
| | C2 | 0.562 | | |
| | C3 | 0.905 | | |
| Extraversion | | | 0.661 | 0.502 |
| | E1 | 0.821 | | |
| | E2 | 0.574 | | |
| Openness to Experience | | | 0.805 | 0.581 |
| | OE1 | 0.773 | | |
| | OE2 | 0.806 | | |
| | OE3 | 0.703 | | |
| Neuroticism | | | 0.748 | 0.603 |
| | N1 | 0.650 | | |
| | N2 | 0.885 | | |
| Recycling Intention (E-waste) | | | 0.881 | 0.651 |
| | R1 | 0.715 | | |
| | R2 | 0.770 | | |
| | R3 | 0.883 | | |
| | R4 | 0.849 | | |

and positive influence on e-waste recycling intentions. As a result, H1, H2, and H3 are acceptable. The results suggest that openness to experience ($\beta = 0.232$; t-value = 4.567; p-value$\leq$ 0.004) significantly influences e-waste recycling intentions, supporting H7. Conversely, agreeableness ($\beta = 0.041$; t-value = 0.682; p-value$\leq$ 0.495), conscientiousness ($\beta = 0.125$; t-

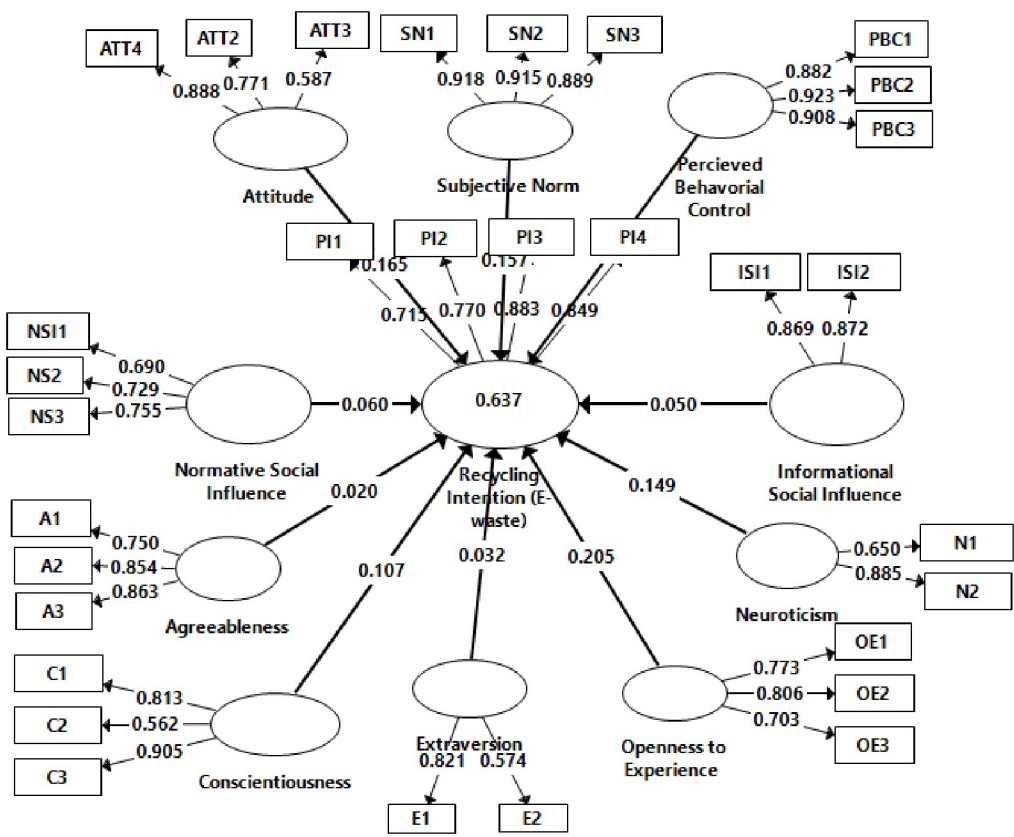

**Fig 2. Measurement model.**

value = 1.814; p-value≤ 0.070), extraversion (β = 0.027; t-value = 0.711; p-value≤ 0.477) and neuroticism (β = 0.016; t-value = 0.264; p-value≤ 0.792) have no significant influence on recycling intention of e-waste. Hence, H4, H5, H6, and H8 are not supported. Our study found that normative social influence (β = 0.091; t-value = 2.399; p-value≤ 0.016) and informational

**Table 3. Discriminant validity.**

|  | AG | AT | C0 | EXT | ISI | NE | NSI | OTE | PBC | RI | SN |
|---|---|---|---|---|---|---|---|---|---|---|---|
| Agreeableness | 0.824 |  |  |  |  |  |  |  |  |  |  |
| Attitude | 0.284 | 0.759 |  |  |  |  |  |  |  |  |  |
| Conscientiousness | 0.446 | 0.647 | 0.774 |  |  |  |  |  |  |  |  |
| Extraversion | 0.098 | 0.043 | 0.107 | 0.708 |  |  |  |  |  |  |  |
| Informational Social Influence | 0.414 | 0.618 | 0.675 | 0.063 | 0.836 |  |  |  |  |  |  |
| Neuroticism | 0.680 | 0.394 | 0.543 | 0.066 | 0.476 | 0.776 |  |  |  |  |  |
| Normative Social Influence | 0.518 | 0.380 | 0.648 | 0.127 | 0.467 | 0.589 | 0.725 |  |  |  |  |
| Openness to experience | 0.322 | 0.572 | 0.583 | 0.065 | 0.559 | 0.391 | 0.564 | 0.762 |  |  |  |
| Perceived Behavioral Control | 0.211 | 0.531 | 0.628 | 0.069 | 0.497 | 0.392 | 0.520 | 0.429 | 0.905 |  |  |
| Recycling Intention (E-waste) | 0.383 | 0.616 | 0.669 | 0.097 | 0.550 | 0.537 | 0.580 | 0.645 | 0.596 | 0.807 |  |
| Subjective Norms | 0.099 | 0.463 | 0.534 | 0.018 | 0.390 | 0.351 | 0.463 | 0.598 | 0.584 | 0.599 | 0.907 |

Note: AG = Agreeableness; At = Attitude; CO = Conscientiousness; EXT = Extraversion; ISI = Informational Social Influence; NE = Neuroticism; NSI = Normative Social Influence; OTE = Openness to experience; PBC = Perceived Behavioral Control; RI = Recycling Intention (E-waste); SN = Subjective Norms

**Table 4. Hypothesis relationship.**

| Hypothetical Relationships | Beta | T-value | P Values | F Values | Status |
|---|---|---|---|---|---|
| H1: Attitude -> Recycling Intention (E-waste) | 0.141 | 3.167 | 0.002 | 0.026 | Accepted |
| H2: Subjective Norms -> Recycling Intention (E-waste) | 0.147 | 2.725 | 0.006 | 0.028 | Accepted |
| H3: Perceived Behavioral Control -> Recycling Intention (E-waste | 0.150 | 2.814 | 0.004 | 0.029 | Accepted |
| H4: Agreeableness -> Recycling Intention (E-waste) | 0.041 | 0.682 | 0.495 | 0.002 | Rejected |
| H5: Conscientiousness -> Recycling Intention (E-waste) | 0.125 | 1.814 | 0.070 | 0.015 | Rejected |
| H6: Extraversion -> Recycling Intention (E-waste) | 0.027 | 0.711 | 0.477 | 0.002 | Rejected |
| H7: Openness to experience -> Recycling Intention (E-waste) | 0.232 | 4.567 | 0.004 | 0.069 | Accepted |
| H8: Neuroticism -> Recycling Intention (E-waste) | 0.016 | 0.264 | 0.792 | 0.000 | Rejected |
| H9: Normative Social Influence -> Recycling Intention (E-waste) | 0.091 | 2.399 | 0.016 | 0.028 | Accepted |
| H10: Informational Social Influence -> Recycling Intention (E-waste) | 0.161 | 2.760 | 0.006 | 0.021 | Accepted |

social influence ($\beta$ = 0.161; t-value = 2.760; p-value$\leq$ 0.006) have a significant relationship to e-waste recycling intentions.

In addition, we consider Cohen's [118] effect size measures of 0.02, 0.15, and 0.35 for small, medium, and large effects. As revealed in Table 4, attitude, subjective norms, perceived behavioral control, openness to experience, informational social influence, and normative social influence value exceeded the 0.02 threshold, signifying a small to medium effect. In contrast, the personality traits agreeableness, conscientiousness, extraversion, and neuroticism had no statistical significance on e-waste recycling intentions.

## 5. Discussion

This study adds new insights into the adoption of recycling intention of e-waste and confirms some previous findings. This study investigates how integrating TPB, personality traits, and social influence theory influences the intention to recycle e-waste among generation Y.

The findings validate the theory of planned behavior by demonstrating a substantial positive link between TPB attitudinal components, subjective norms, perceived behavioral control, and e-waste recycling intention. According to our findings, attitude influences millennials' inclination to recycle e-waste. These findings were constant with previous research [33, 119, 120]. The results indicate that attitude is a strong motivator for Lithuanian consumers of the Y generation to recycle e-waste. It suggests that millennials in Lithuania know well about e-waste and how it affects the environment and human health. The TPB predictors' subjective norms positively influence consumers' e-waste recycling intention, consistent with previous research [33, 121]. Even though Lithuania is an individualistic culture, the results reveal that millennials feel pressure from important others (e.g., friends, coworkers, media, and society) to engage in e-waste recycling activities. Our findings show that perceived behavioral control influences millennials' intentions to recycle e-waste. The findings converge with recent research that found that PBC favors recycling intentions [119, 122–124]. As a result, Millennials in Lithuania are more persuaded to recycle e-waste and are prepared to devote their time and energy. Additionally, millennials find it easier to dispose of their e-waste in this way (better financial value, door-to-door collection, online sale).

Agreeableness has a non-significant influence on millennials' e-waste recycling intention. The findings contradict past studies that agreeableness positively correlates with environmentally-friendly intentions and behavior [42, 86, 87]. The results suggest that millennials in Lithuania are individualistic and self-centered, are unwilling to prioritize the interests of others, and are unlikely to participate in eco-friendly activities such as recycling e-waste. Likewise,

conscientiousness has a non-significant influence on the recycling intention of e-waste. Again, the findings contradict past studies [45, 86, 87]. More specifically, the results suggest that the facets of conscientiousness (self-discipline, a sense of duty, obligations, and adherence to rules and customs) do not encourage consumers' intention to recycle e-waste in the context of Lithuania. In addition, our study found that extraversion has a non-significant relationship with e-waste recycling intentions. The findings parallel past studies in that extraversion has a non-significant association with pro-environmental behavior [43, 99]. Duong [42] also emphasized that extraversion is less essential in sustainable consumption. This discrepancy can be explained by the fact that most previous research on pro-environmental behavior has concluded that the relationship between extraversion and pro-environmental behavior is often insignificant [43, 45]. From all the personality traits investigated in this study, only openness to experience is significantly related to e-waste recycling intentions. More specifically, the study found that openness to experience significantly affects millennials' e-waste recycling intentions, which contradicts almost all previous studies [86, 89, 90]. The result suggests that millennial consumers in Lithuania are creative, innovative, and eco-friendly; therefore, millennials do not hesitate to embrace novelty and adapt to unusual experiences such as e-waste recycling. Bhutto et al. [64] comment that Generation Y individuals face more financial constraints than previous generations, though they are motivated toward sustainable consumption. Our study found that neuroticism has a non-significant impact on Y-generation consumers' e-waste recycling intentions in Lithuania. These findings support several past studies [42, 89]. Therefore, results indicated that concerned and easily upset consumers might be reluctant to adopt sustainable consumption practices like e-waste recycling.

In contrast to personality traits, social influence theory constructs seem more prospective to predict e-waste recycling intentions. Normative social influence positively influences the recycling intention of e-waste, and these findings support past studies on recycling behavior [48, 55, 104]. This result suggests that the normative social influences experienced by Y-gen consumers (i.e., Millennials) inspire them to intend to recycle e-waste. Similarly, informational social influence significantly influences the recycling intention of e-waste, a finding that supports past studies on recycling behavior [48, 125]. Millennials receiving more information about e-waste recycling are more likely to intend to implement the conduct for which they have received more information. They sense they are doing the right by recycling e-waste because others also reinforce their belief in e-waste recycling.

## 6. Conclusion

### Theoretical contributions

Our findings have significant theoretical and practical implications. From a theoretical point of view, this research has three major contributions. Firstly, E-waste recycling behavior is well elucidated by preceding literature [34, 35, 38, 82, 126, 127]; this study extends the current knowledge by integrating TPB, social influence theory, and personality traits to examine consumer recycling behavior towards e-waste.

Second, lack of literature on the general acceptance of e-waste recycling behavior, especially among Generation Y [33]. Although climate change significantly impacts Generation Y, it has been neglected in environmental studies. This study is unique because it includes Generation Y, a population segment that remains primarily understudied in sustainability research despite its growing importance and perspective.

Thirdly, recycling behavior is well expounded by preceding literature in the context of different countries [14–16, 24, 25, 27, 127, 128], but research on e-waste recycling in the context of Baltic countries is scant. Lithuania is a potentially promising market for sustainable

consumption due to its environmentally-minded population and growing economy. This study aimed to gain deeper insights into consumers' e-waste recycling intentions in the Baltic market (Lithuania).

## Practical and managerial contributions

From a practical point of view, this research is expected to help guide administrative practice in various ways. First, the results of this study are a significant contribution to the sustainability literature, especially for the electronics industry in Lithuania. Based on the findings, our study suggests attitude, subjective norms, and perceived behavioral control encourage generation Y e-waste recycling intention. Marketers should promote the environmental and social benefits of e-waste recycling, such as reduced mining of scarce resources (gold, palladium, copper, and silver); electronic recycling promotes the integrity of agricultural soil and reduces health hazards on the environment.

Secondly, the results of this study suggest managers and policymakers should consider personality traits in their campaigns and policies to encourage e-waste recycling behaviors. Policymakers and managers must know consumer personality traits to propose environmentally sustainable measures. In particular, policymakers should have appropriate solutions to promote sustainable consumption behavior, protect society's environment, and contribute to sustainable development.

Third, the study suggests that society's influence, whether through informational or normative social influence, drives consumers to engage in e-waste recycling activities. The government and managers should spread positive messages about e-waste through various media platforms. As society manipulates the behavior of individuals, community engagement programs can be initiated. Policymakers are also advised to consider the role of family and peers when developing communication strategies. Social media is also an effective technique to attract them. Managers and companies could also use social media platforms (like WhatsApp, Facebook, Instagram, and Twitter) to spread information about e-waste recycling and promote sustainable behaviors. Not only is this strategy cost-effective, but it can also attract most consumers.

## Limitations and future research directions

Despite its unique contribution, this study has limitations. First, this study takes place in Lithuania; it incorporates the TPB, social influence theory, and personality factors, integrating or merging the motives for the long-term acceptance of e-waste. Because this study focuses on consumer behavior regarding e-waste acceptance, the findings cannot be applied to other contexts, such as solid waste or garment disposal. Second, we concentrate on e-waste recycling intention. Future studies on actual behavior may compare intent and behavior to better understand how intentions convert into actual behavior. Third, e-waste is a substantial issue in the Baltic nations, dealing with an exponentially increasing e-waste problem. As a result, while the technique selected by this study may be broadly extended outside Lithuania's geography, its findings' generalizability should be approached with care in the context of Baltic nations with cultural and social values that differ from Lithuania's.

## Supporting information

**S1 Appendix.**
(DOCX)

## Author Contributions

**Conceptualization:** Muhammad Yaseen Bhutto, Aušra Rūtelionė.

**Data curation:** Muhammad Yaseen Bhutto, Aušra Rūtelionė.

**Formal analysis:** Muhammad Yaseen Bhutto, Aušra Rūtelionė, Myriam Ertz.

**Funding acquisition:** Beata Šeinauskienė.

**Methodology:** Muhammad Yaseen Bhutto, Aušra Rūtelionė, Myriam Ertz.

**Project administration:** Beata Šeinauskienė.

**Software:** Muhammad Yaseen Bhutto, Aušra Rūtelionė, Beata Šeinauskienė.

**Supervision:** Beata Šeinauskienė.

**Validation:** Muhammad Yaseen Bhutto, Aušra Rūtelionė, Beata Šeinauskienė, Myriam Ertz.

**Visualization:** Myriam Ertz.

**Writing – original draft:** Muhammad Yaseen Bhutto, Aušra Rūtelionė.

**Writing – review & editing:** Muhammad Yaseen Bhutto, Aušra Rūtelionė, Beata Šeinauskienė, Myriam Ertz.

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
