## [Decision Letter · Decision Letter 0]

6 Mar 2023

PONE-D-22-33671Exploring factors of e-waste recycling intention:  the case of generation Y in LithuaniaPLOS ONE

Dear Muhammad Yaseen Bhutto Bhutto,

Thank you for submitting your manuscript to PLOS ONE. After careful consideration, we feel that it has merit but does not fully meet PLOS ONE’s publication criteria as it currently stands. Therefore, we invite you to submit a revised version of the manuscript that addresses the points raised during the review process.

We look forward to receiving your revised manuscript.

Kind regards,

Tai Ming Wut

Academic Editor

PLOS ONE

Journal Requirements:

The research is part of the project “CD-TOOLS. CD TOOLS for product integrity” no.: 01.2.2-LMT-K-718-03-0104, funded by the European Regional Development Fund according to the 2014–2020 Operational Programme for the European Union Funds’ Investments, under measure’s Sustainability 2022, 14, 5021 16 of 18 No. 01.2.2-LMT-K-718 activity “Research Projects Implemented by World-class Researcher Groups to develop R&D activities relevant to economic sectors, which could later be commercialized”

Reviewers' comments:

Reviewer's Responses to Questions

**Comments to the Author**

1. Is the manuscript technically sound, and do the data support the conclusions?

Reviewer #1: Yes

Reviewer #2: Yes

2. Has the statistical analysis been performed appropriately and rigorously? 

Reviewer #1: Yes

Reviewer #2: Yes

3. Have the authors made all data underlying the findings in their manuscript fully available?

Reviewer #1: Yes

Reviewer #2: Yes

4. Is the manuscript presented in an intelligible fashion and written in standard English?

Reviewer #1: Yes

Reviewer #2: Yes

5. Review Comments to the Author

Reviewer #1: The topic is very interesting, and I enjoyed it. I would like to thank you for your efforts in presenting your research work in such a professional manner. there are some inspiring insights thorough the manuscript, and I tend to agree on its publication. However, before your work is recommended or accepted, a few comments must be included to improve the quality of your work as well as for future publication in this reputable journal.

I would like to see the revised manuscript. I encourage authors to carefully consider and incorporate the suggested articles for the manuscript quality improvements and justify where necessary. While the suggested citations have been made in the spirit of covering the existing and relevant literature for the author information. Authors feels free to cite suggested citations for the manuscript quality.

The authors must modify the following points in great detail.

-I have the following observations, questions, and comments that may help to improve your work. However, there are few points that needs to be quickly addressed to improve its overall communication:

In the abstract, introduction (urgency and significance of the research hypothesis. Also please include methodology and managerial implications from the findings of this study in the context of the environment by combining the research objectives and problems.

The introduction section needs a few more sentences to strengthen the article, and please include the research problem, objective, and novelty in the last paragraph of the Introduction section. Include a few more sentences at the beginning of the introduction explaining your paper's contribution, as well as your attempts to deal with or present solutions to a specific problem/s and your unique contribution with this research paper. I suggest the following articles.

Integrating knowledge management and orientation dynamics for organization transition from eco-innovation to circular economy, Journal of Knowledge Management, https://doi.org/10.1108/JKM-05-2022-0424

Designing Value Chains for Industry 4.0 and a Circular Economy: A Review of the Literature. Sustainability. 2022; 14(12):7084. https://doi.org/10.3390/su14127084

Environmental Effects of Bio-Waste Recycling on Industrial Circular Economy and Eco-Sustainability. Recycling. 2022; 7(4):60. https://doi.org/10.3390/recycling7040060

-Furthr, though your study is very relevant in the present times, the paper has not been developed adequately in terms of structure, operationalization of variables, justification of the scales adopted, and explanation of the results. It is important to highlight what is missing/unclear in literature and how your study fills gaps/clarifies; what are the wider implications. The introduction is very ambiguous and does not clearly establish the need, purpose and contribution of this study. The specific objectives of the study are not clearly listed by the authors.

-In the conclusion, clearly condensate the novelty and significance of the main discovery into a short and ground-breaking claim. Highlight the significance of your conclusions (how have the boundaries of human knowledge been expanded?) and clearly indicate how will our readers benefit from these findings (explain the applicability).

-Please understand that the Conclusion chapter is not a summary of your work, present only original and industrially significant revelations that have the potential to expand the horizon of human knowledge (higher level of generalization is mandatory. In terms of the flow and structure of the paper, better attention could have been given to developing the background of the study and establishing why this topic is important for research.

- I suggest the authors refine the managerial insights based on the findings. The writing of the paper needs a lot of improvement in terms of grammar, spelling, and presentations. he various variables used in the study could have been operationalized better. The findings have only focused on the statistical significance or lack of the same and not the underpinnings of the same.

Reviewer #2: The study highlights the need to investigate consumers' behavioural intentions towards e-waste recycling, particularly among millennials, who are the largest segment of consumers of electronic products. By integrating the theory of planned behaviour, social influence theory, and personality traits, the study identifies significant factors that influence consumers' e-waste recycling intention. The results indicate that attitude, subjective norms, and perceived behavioural control significantly influence consumers' e-waste recycling intention. Additionally, openness to experience was found to be the only personality trait that significantly affects consumers' e-waste recycling intention. The study's findings have important implications for policymakers and marketers in understanding and encouraging e-waste behaviour among Lithuanian Y-generation consumers.

Here are some suggestions for the manuscripts:

1. Please provide highlighted and clearer research questions directly that can help the reader understand the research purpose easily. What is the problem you are trying to solve? What is the significance of the research? What can the research contribute to the actual application? Please include these in the introduction.

2. The review section has been done logically. However, it lacks a short summarization of TPB, especially on e-waste behaviour. Also, people's personalities are often complex, the section missed some discussion of diverse personalities. Some articles for information:

Pathways of place dependence and place identity influencing recycling in the extended theory of planned behavior. Journal of Environmental Psychology, 81, 101795.

Exploring the heterogeneity in drivers of energy-saving behaviours among hotel guests: Insights from the theory of planned behaviour and personality profiles. Environmental Impact Assessment Review, 99, 107012.

3. Can the sample represent the overall situation of the country? e.g. Can the sample of over 60% of respondents with a university degree represent the education level of the country? A table of demography information can help to identify the sample.

4. Is the proposed model has a good model fit?

5. Some presented values in measurement model (figure 2 and table 1) are difference, such as Neuroticism, Informational Social Influence, better double check on the presented data and items.

6. Clarify the findings. Please provide a clearer summary of your findings. What are the main results of your study? What are the implications of your findings for policymakers and marketers? Please explain these in the conclusion section.

6. PLOS authors have the option to publish the peer review history of their article (what does this mean?). If published, this will include your full peer review and any attached files.

Reviewer #1: No

Reviewer #2: No

---

## [Author Response · Author response to Decision Letter 0]

10 Apr 2023

Dear Editorial Team and Reviewers,

We appreciate the comments and suggestions made by the review and editorial team. Indeed, we think that our work has substantially improved due to these comments. We have used the track changes function in Word to highlight our modifications. In addition, our answers to the reviewers’ comments below provide additional detail and explanations as to what has been done specifically to improve the quality of the paper.

Reviewer 1:

Comment 1: The topic is very interesting, and I enjoyed it. I would like to thank you for your efforts in presenting your research work in such a professional manner. there are some inspiring insights thorough the manuscript, and I tend to agree on its publication. However, before your work is recommended or accepted, a few comments must be included to improve the quality of your work as well as for future publication in this reputable journal. I would like to see the revised manuscript. I encourage authors to carefully consider and incorporate the suggested articles for the manuscript quality improvements and justify where necessary. While the suggested citations have been made in the spirit of covering the existing and relevant literature for the author information. Authors feels free to cite suggested citations for the manuscript quality.

Response 1: Thanks for reviewing our manuscript and encouraging comments. They have helped us to craft what we believe is a significantly improved version of our paper.

Comment 2: The authors must modify the following points in great detail.

-I have the following observations, questions, and comments that may help to improve your work. However, there are few points that needs to be quickly addressed to improve its overall communication. In the abstract, introduction (urgency and significance of the research hypothesis. Also please include methodology and managerial implications from the findings of this study in the context of the environment by combining the research objectives and problems.

Response 2: Thanks for the comment. As rightfully suggested by the reviewer, the abstract and introduction parts have been improved accordingly. More specifically, the abstract also states the methodology and managerial implications, while the urgency and significance of the research have been better articulated in the first paragraph of the Introduction.

Comment 3: The introduction section needs a few more sentences to strengthen the article, and please include the research problem, objective, and novelty in the last paragraph of the Introduction section. Include a few more sentences at the beginning of the introduction explaining your paper's contribution, as well as your attempts to deal with or present solutions to a specific problem/s and your unique contribution with this research paper. I suggest the following articles.

Integrating knowledge management and orientation dynamics for organization transition from eco-innovation to circular economy, Journal of Knowledge Management, https://doi.org/10.1108/JKM-05-2022-0424

Designing Value Chains for Industry 4.0 and a Circular Economy: A Review of the Literature. Sustainability. 2022; 14(12):7084. https://doi.org/10.3390/su14127084

Environmental Effects of Bio-Waste Recycling on Industrial Circular Economy and Eco-Sustainability. Recycling. 2022; 7(4):60. https://doi.org/10.3390/recycling7040060

Response 3: The introduction has been improved, as suggested. First, we have included the research problem, objective, novelty, and research questions in the last paragraph. Second, the fourth and fifth paragraphs have been revised to emphasize better the paper’s contribution and our attempts to deal with or present solutions to the specific problem of promoting e-waste recycling behavior among consumers. Third, we thank the reviewer for the insightful suggested papers. We have cited both “Designing Value Chains for Industry 4.0 and a Circular Economy” and “Environmental Effects of Bio-Waste Recycling on Industrial Circular Economy and Eco-Sustainability” in the article because these two studies are more suitable to the research topic. 

Comment 4: Furthr, though your study is very relevant in the present times, the paper has not been developed adequately in terms of structure, operationalization of variables, justification of the scales adopted, and explanation of the results. It is important to highlight what is missing/unclear in literature and how your study fills gaps/clarifies; what are the wider implications. The introduction is very ambiguous and does not clearly establish the need, purpose and contribution of this study. The specific objectives of the study are not clearly listed by the authors.

Response 4: Thanks for the comment. We have revised the introduction, taken special care in clarifying the research gap and objectives, and better explained the constructs. Moreover, the operationalization of the variables is specified in sub-section 3.2. Measures while Appendix 1 lists all the items used to measure the different studied constructs. We kindly refer the reviewer to the Introduction, sub-section 3.2, and Appendix 1 for more details.

Comment 5: In the conclusion, clearly condensate the novelty and significance of the major discovery into a short and ground-breaking claim. Highlight the significance of your conclusions (how have the boundaries of human knowledge been expanded?) and clearly indicate how will our readers benefit from these findings (explain the applicability). -Please understand that the Conclusion chapter is not a summary of your work, present only original and industrially significant revelations that have the potential to expand the horizon of human knowledge (higher level of generalization is mandatory. In terms of the flow and structure of the paper, better attention could have been given to developing the background of the study and establishing why this topic is important for research.

Response 5: Thanks for the comment, and by the conclusion, we understand that the reviewer refers to the last section, 5. The discussion, which we have renamed 5. Discussion and conclusion. As suggested, we have revised the whole section by setting the discussion of the results apart from the theoretical contributions. The latter have also been reformulated to better condense our findings' novelty and significance in three main paragraphs, each corresponding to a specific short and ground-breaking claim. Furthermore, while this part provides some background about the problem, it underscores the urgency and cruciality of the research. The practical and managerial contributions, limitations, and future research directions are discussed in a third and fourth sub-part.

Comment 6: I suggest the authors refine the managerial insights based on the findings. The writing of the paper needs a lot of improvement in terms of grammar, spelling, and presentations. he various variables used in the study could have been operationalized better. The findings have only focused on the statistical significance or lack of the same and not the underpinnings of the same.

Response 6: The managerial contributions sub-section has been completely revised for better presentation, flow, and accuracy. We took a higher level of analysis by considering the results from a managerial perspective rather than a mere statistical one. We kindly refer the reviewer to sub-section 5.3 for more details. 

Reviewer 2 

Comment 1: The study highlights the need to investigate consumers' behavioural intentions towards e-waste recycling, particularly among millennials, who are the largest segment of consumers of electronic products. By integrating the theory of planned behaviour, social influence theory, and personality traits, the study identifies significant factors that influence consumers' e-waste recycling intention. The results indicate that attitude, subjective norms, and perceived behavioural control significantly influence consumers' e-waste recycling intention. Additionally, openness to experience was found to be the only personality trait that significantly affects consumers' e-waste recycling intention. The study's findings have important implications for policymakers and marketers in understanding and encouraging e-waste behaviour among Lithuanian Y-generation consumers.

Comment 1: First, we would like to thank you for taking the time to read our paper and provide valuable comments for its improvement. We have considered each of those comments and subsequently explain how we addressed them point by point. 

Comment 2: Please provide highlighted and clearer research questions directly that can help the reader understand the research purpose easily. What is the problem you are trying to solve? What is the significance of the research? What can the research contribute to the actual application? Please include these in the introduction.

Response 2: The Introduction has been significantly revised by mentioning the research, objective, novelty, and research questions in the last paragraph. In addition, the fourth and fifth paragraphs have been revised to better emphasize the paper’s contribution and our attempts to deal with or present solutions to the specific problem of promoting e-waste recycling behaviour among consumers. This shows the actual application to which this research contributes.

Comment 3: The review section has been done logically. However, it lacks a short summarization of TPB, especially on e-waste behaviour. Also, people's personalities are often complex, the section missed some discussion of diverse personalities. Some articles for information:

Pathways of place dependence and place identity influencing recycling in the extended theory of planned behavior. Journal of Environmental Psychology, 81, 101795.

Exploring the heterogeneity in drivers of energy-saving behaviours among hotel guests: Insights from the theory of planned behaviour and personality profiles. Environmental Impact Assessment Review, 99, 107012.

Response 3: As suggested, we have completely revised the theoretical framework sub-section, which deals with the theory of planned behaviour (TPB). We notably better highlighted the importance of the TPB, and we have also summarized the e-waste behavior. We have also cited your suggested papers in our manuscript. In addition, regarding the discussion about the diverse personalities, we thought that this would best fit in the Introduction section because it relates to a specific model variable rather than to the theoretical framework itself. Therefore, we also mention the EIAR paper in that discussion. The changes can be found in sub-section 2.1 and in the fourth paragraph of the Introduction section. 

Comment 4: Can the sample represent the overall situation of the country? e.g. Can the sample of over 60% of respondents with a university degree represent the education level of the country? A table of demography information can help to identify the sample.

Response 4: As surprising as it may seem, this figure is quite representative of the Lithuanian population. In fact, about two-thirds of the Lithuanians in their thirties have a university degree which is more than in many European countries and other Baltic states such as Latvia or Estonia, where the share is already high at 40% (MOSTA). It fares way better than the OECD average in this regard. Besides, Lithuania has the highest share of students under 29 years old. Considering the fact that our sample mainly comprises respondents in their 20s or their 30s, the high percentage of respondents with at least one university degree is thus not so surprising. As requested, we have incorporated a demographic table for better understanding. 

Comment 5: Is the proposed model has a good model fit?

Response 5: Our PLS-SEM analytical approach does not use conventional fit statistics like NFI, CFI, or even chi-square since it is based on partial least squares modeling. Consequently, this explains the absence of such indices if this is what the reviewer was initially looking for. However, alternative indicators are used, such as the R-square value. In our model, it amounts to 0.630, which means that the model accounts for more than half of the variance in the dependent variable, which is relatively high, especially in the context of the social sciences. 

Comment 6: Some presented values in measurement model (figure 2 and table 1) are difference, such as Neuroticism, Informational Social Influence, better double check on the presented data and items. Clarify the findings. Please provide a clearer summary of your findings. What are the main results of your study? What are the implications of your findings for policymakers and marketers? Please explain these in the conclusion section.

Response 6: We have double-checked the values in the measurement model table and in the figure for better clarity and accuracy. In addition, the conclusive portion of the paper has been substantially revised by clearly delineating the discussion of the results from the theoretical implications and the managerial implications for policymakers and marketers.

---

## [Decision Letter · Decision Letter 1]

7 May 2023

PONE-D-22-33671R1Exploring factors of e-waste recycling intention:  the case of generation Y in LithuaniaPLOS ONE

Dear Dr. Bhutto,

Thank you for submitting your manuscript to PLOS ONE. After careful consideration, we feel that it has merit but does not fully meet PLOS ONE’s publication criteria as it currently stands. Therefore, we invite you to submit a revised version of the manuscript that addresses the points raised during the review process.

We look forward to receiving your revised manuscript.

Kind regards,

Tai Ming Wut

Academic Editor

PLOS ONE

Journal Requirements:

Reviewers' comments:

Reviewer's Responses to Questions

**Comments to the Author**

1. If the authors have adequately addressed your comments raised in a previous round of review and you feel that this manuscript is now acceptable for publication, you may indicate that here to bypass the “Comments to the Author” section, enter your conflict of interest statement in the “Confidential to Editor” section, and submit your "Accept" recommendation.

Reviewer #2: (No Response)

2. Is the manuscript technically sound, and do the data support the conclusions?

Reviewer #2: Yes

3. Has the statistical analysis been performed appropriately and rigorously? 

Reviewer #2: Yes

4. Have the authors made all data underlying the findings in their manuscript fully available?

Reviewer #2: Yes

5. Is the manuscript presented in an intelligible fashion and written in standard English?

Reviewer #2: Yes

6. Review Comments to the Author

Reviewer #2: This study presents a valuable contribution to understanding e-waste recycling behaviour among Lithuanian millennials. The use of the PLS-SEM approach to analyze the data is appropriate and provides robust results. The findings of the study, particularly the significant influence of attitude, subjective norms, and perceived behavioural control on consumers' e-waste recycling intention, have important implications for policymakers.

Most of the comments were addressed well. However, the previous question of model fit is not that clear.

1. Some papers (Determinants and mechanisms driving energy-saving behaviours of long-stay hotel guests: Comparison of leisure, business and extended-stay residential cases), (the EIAR) as well as (the JEP) had described it, the author could consider explaining more about the R square in your manuscript to help readers to clearly identify the valuable and meaningful of this value.

2. Besides, it seems there is an inconsistency of the R2 value in Figure 2 (0.637) and text (0.630), please kindly check on the result.

3. In addition, indeed, it is some studies used the indices of R square to evaluate the model, but the value of SRMR is usually used to assess the fit of the model for the PLS-SEM method. Just a suggestion, the author can consider using second indices to discuss the fit if it is appropriate.

7. PLOS authors have the option to publish the peer review history of their article (what does this mean?). If published, this will include your full peer review and any attached files.

Reviewer #2: No

---

## [Author Response · Author response to Decision Letter 1]

8 May 2023

We appreciate the comments and suggestions made by the review and editorial team. Indeed, we think that our work has substantially improved due to these comments. We have used the track changes function in Word to highlight our modifications. In addition, our answers to the reviewers’ comments below provide additional detail and explanations as to what has been done specifically to improve the quality of the paper.

1. Reviewer Comment: This study presents a valuable contribution to understanding e-waste recycling behaviour among Lithuanian millennials. The use of the PLS-SEM approach to analyze the data is appropriate and provides robust results. The findings of the study, particularly the significant influence of attitude, subjective norms, and perceived behavioural control on consumers' e-waste recycling intention, have important implications for policymakers.

Most of the comments were addressed well. However, the previous question of model fit is not that clear.

Authors Response: I have mentioned the model indicators in the manuscript. I also mentioned the inner model values including saturated and estimated model values. Kindly check the data analysis part. Thanks 

2. Reviewer comment: Some papers (Determinants and mechanisms driving energy-saving behaviours of long-stay hotel guests: Comparison of leisure, business and extended-stay residential cases), (the EIAR) as well as (the JEP) had described it, the author could consider explaining more about the R square in your manuscript to help readers to clearly identify the valuable and meaningful of this value.

Authors Response: I have incorporated further explanation of R saquare with literature support in the manuscript.

3. Reviewer comment: Besides, it seems there is an inconsistency of the R2 value in Figure 2 (0.637) and text (0.630), please kindly check on the result.

Authors Response: I have incorporated and corrected in the manuscript. 

4. In addition, indeed, it is some studies used the indices of R square to evaluate the model, but the value of SRMR is usually used to assess the fit of the model for the PLS-SEM method. Just a suggestion, the author can consider using second indices to discuss the fit if it is appropriate.

Authors Response: Thanks for suggestion. Suggestion has been incorporated.

---

## [Editor Report · Decision Letter 2]

29 May 2023

PONE-D-22-33671R2Exploring factors of e-waste recycling intention:  the case of generation Y in LithuaniaPLOS ONE

Dear Muhammad Yaseen Bhutto Bhutto,

Thank you for submitting your manuscript to PLOS ONE. After careful consideration, we feel that it has merit but does not fully meet PLOS ONE’s publication criteria as it currently stands. Therefore, we invite you to submit a revised version of the manuscript that addresses the points raised during the review process.

We look forward to receiving your revised manuscript.

Kind regards,

Tai Ming Wut

Academic Editor

PLOS ONE

Journal Requirements:

Additional Editor Comments:

Please add a conclusion at the end.

---

## [Author Response · Author response to Decision Letter 2]

30 May 2023

Dear Editor and Reviewers, 

 Thanks for your precious comments and time. I have revised the manuscript references and In text citation according to the journal requirements. I also separated the discussion and conclusion part.

---

## [Editor Report · Decision Letter 3]

7 Jun 2023

Exploring factors of e-waste recycling intention:  the case of generation Y in Lithuania

PONE-D-22-33671R3

Dear Muhammad Yaseen Bhutto Bhutto,

We’re pleased to inform you that your manuscript has been judged scientifically suitable for publication and will be formally accepted for publication once it meets all outstanding technical requirements.

Kind regards,

Tai Ming Wut

Academic Editor

PLOS ONE
---

## [Editor Report · Acceptance letter]

16 Jun 2023

PONE-D-22-33671R3 

Exploring factors of e-waste recycling intention:  the case of generation Y in Lithuania 

Dear Dr. Bhutto:

I'm pleased to inform you that your manuscript has been deemed suitable for publication in PLOS ONE. Congratulations! Your manuscript is now with our production department. 

Kind regards, 

on behalf of

Dr. Tai Ming Wut 

Academic Editor

PLOS ONE